# Acceptance of a Nordic, Protein-Reduced Diet for Young Children during Complementary Feeding—A Randomized Controlled Trial

**DOI:** 10.3390/foods10020275

**Published:** 2021-01-29

**Authors:** Ulrica Johansson, Lene Lindberg, Inger Öhlund, Olle Hernell, Bo Lönnerdal, Saara Lundén, Mari Sandell, Torbjörn Lind

**Affiliations:** 1Department of Clinical Sciences, Pediatrics, Umeå University, SE 901 85 Umeå, Sweden; inger.ohlund@umu.se (I.Ö.); olle.hernell@umu.se (O.H.); torbjorn.lind@umu.se (T.L.); 2Department of Public Health Sciences, Karolinska Institutet and Centre for Epidemiology and Community Medicine, Stockholm County Council, SE 104 31 Stockholm, Sweden; lene.lindberg@ki.se; 3Department of Nutrition, University of California, Davis, CA 95616, USA; bllonnerdal@ucdavis.edu; 4Functional Foods Forum, University of Turku, FI-20014 Turku, Finland; saara.lunden@utu.fi (S.L.); mari.sandell@helsinki.fi (M.S.); 5Department of Food and Nutrition, University of Helsinki, FI-00014 Helsinki, Finland

**Keywords:** infant feeding, healthy eating, food preference, eating behavior, repeated exposure, vegetables, fruits, sustainable eating, environment

## Abstract

Early life is critical for developing healthy eating patterns. This study aimed to investigate the effects of a Nordic, protein-reduced complementary diet (ND) compared to a diet following the current Swedish dietary guidelines on eating patterns and food acceptance. At 4–6 months (mo) of age infants were randomized to a Nordic group (NG, *n* = 41) or a Conventional group (CG, *n* = 40), and followed until 18 mo of age. Daily intake of fruits and vegetables (mean ± sd) at 12 mo was significantly higher in the NG compared to the CG: 341 ± 108 g/day vs. 220 ± 76 g/day (*p* < 0.001), respectively. From 12 to 18 mo, fruit and vegetable intake decreased, but the NG still consumed 32% more compared to the CG: 254 ± 99 g/day vs. 193 ± 67 g/day (*p* = 0.004). To assess food acceptance, both groups were tested with home exposure meals at 12 and 18 mo. No group differences in acceptance were found. We find that a ND with parental education initiates healthy eating patterns during infancy, but that the exposure meal used in the present study was insufficient to detect major differences in food acceptance. This is most likely explained by the preparation of the meal. Nordic produce offers high environmental sustainability and favorable taste composition to establish healthy food preferences during this sensitive period of early life.

## 1. Introduction

Early nutrition is fundamental to growth and development of children [1]. This sensitive period of early infancy [2,3] also plays an essential role in establishing long-standing eating behaviors and dietary patterns [4,5,6,7,8]. During this limited period, brain plasticity and sensory pathways are receptive to effects of experiences like preferences for different types of food [9,10]. Compared to recommendations, there is an underconsumption of fruits and vegetables among children and adults, which increases the risk of future non-communicable diseases [11,12,13,14,15,16]. It is therefore urgent to find ways to meet the dietary guidelines for fruits and vegetables and to help develop viable dietary patterns, which can also contribute to a sustainable environment [17,18]. To increase children’s fruit and vegetable intake through child-feeding strategies, methods such as repeated exposures and parent nutrition education interventions have been evaluated [19,20,21,22,23]. Most of this research demonstrated short-term impact, lasting less than 12 months without multicomponent interventions and did not include experiences of food prepared and eaten at home, i.e., the setting where most children consume a major part of their daily energy and nutrient intake [24]. There is at present a lack of studies where multicomponent, long-term (i.e., >12 months) dietary interventions during infancy have been combined with measures of child-feeding practices together with parent nutrition education such as cooking skills for homemade baby food and knowledge about healthy, sustainable, regionally produced food [24].

During infancy, a high protein intake has been associated with higher body mass index (BMI) and overweight later in life [25,26,27,28]. In some Nordic countries the protein intake is high during complementary feeding (CF) in comparison to the Nordic nutrition recommendations for infants [29]. When aiming to diminish animal protein intake in order to reduce the risk of later overweight and obesity [30,31,32,33], carbohydrates from fruits and vegetables can be used to allow the meals to remain isocaloric [34]. In the present study, our definition of CF emanates from the one supplied by the World Health Organization (WHO) [35] with the addition of formula feeding. Thus, we define CF as the process when breast-milk and/or formula alone is no longer sufficient to meet the nutritional requirement of infants, and therefore other foods and liquids are needed, along with breast-milk and/or formula.

The ND, which emphasizes more plant-based foods rich in fruits, berries, vegetables, roots, legumes and whole grains, but less foods from animals, has appropriate characteristics to be suitable during CF, also bringing high environmental sustainability [36,37]. Studies on ND among Danish school children [38,39], as well as on adults from the Nordic countries [40,41,42,43], have shown positive health effects. Likewise, high adherence to ND during pregnancy has been associated with optimal weight gain and improved fetal growth [44]. The WHO has concluded that ND has health-promoting properties and efficacy in reducing non-communicable diseases comparable with the Mediterranean diet [45].

Food acceptance and dietary intake among infants can be influenced by factors such as parents’ sensitivity and feeding practices during the meal situation [46,47]) as well as the infant’s own temperament [48,49]. There is a vulnerable transfer period from 1 to 1.5 years of age, when toddlers are changing from baby food to family food [46]. Few studies have examined how eating behavior changes during this period of life [46].

The overall aim of the present study was to investigate if early repeated exposure with Nordic fruits and vegetables in a protein-reduced, Nordic complementary diet have long-term effects (>12 mo) on eating behavior and food acceptance compared to following the current Swedish recommendations for infants until 18 mo of age. More specifically, to explore this we used dietary food records (FR) and consumption patterns between the groups to assess eating behavior and videotaped exposure meals where snacks made of Nordic ingredients were served to the two groups to assess food acceptance. We hypothesized that providing parents with (a) taste portion schedule for repeated exposure at 4–6 mo of age with Nordic, homemade baby food recipes with fruits and vegetables, and (b) protein-reduced baby food products and homemade baby food recipes from 6 to 18 mo of age, and (c) educational support to parents through social media to increase their knowledge of preparing and using Nordic foods in CF, would increase their infants’ intake of Nordic fruits and vegetables and achieve higher acceptance of Nordic healthy foods compared to a group following the current Swedish dietary recommendations for infants.

## 2. Materials and Methods

### 2.1. Study Participants, Allocation and Blinding

Participants in the present study constitute a sub-group of the 250 four to six-month-old infants recruited to the “Optimized complementary feeding study” (OTIS; ClinicalTrials.gov registration number NCT02634749) from April 2015 and January 2018, in Umeå, Sweden. A detailed description of the study protocol has been published elsewhere [50]. In the current study, one-third of the participants in the OTIS study were randomly allocated after study start (Figure 1) to evaluate eating behaviors and food acceptance at 12 and 18 mo of age from both the Nordic group (NG) and Conventional group (CG). A subsample was used due to the extra effort for the families to participate in the home exposure meals and to be cost-efficient. Inclusion and exclusion criteria for this effort were the same as for the OTIS study, i.e., healthy singleton infants, 4–6 mo of age at inclusion, born after >37 weeks of gestation with birth weight >2500 g, and living in and intending to remain in Umeå during the 12 mo duration of the study, and with no chronic illnesses that would affect nutrient intake or the outcomes of the study. Other exclusion criteria were iron deficiency, any other biochemical abnormality or having started CF at the time of recruitment. Infants were exclusively breast-fed and/or formula-fed at the study start. Parents, research nurses and dieticians who calculated the dietary intake from the FR could not be completely blinded to the group allocation since the food items given to the infants could not be blocked entirely. However, the parents were blinded to the hypotheses of the study. The pediatric dietician, who videotaped the exposure meal in the family’s home, and the psychologist and masters students in psychology who coded and analyzed the videotaped exposure meals were blinded for the infant’s group allocation as were the laboratory team and researchers responsible for the laboratory analyses.

### 2.2. New Nordic Food and Diet

The food in the NG contained 100% Nordic ingredients and the selection of food items was created around the concept “New Nordic Food Manifesto” from the Nordic Council of Ministers [51,52], which promotes season-based Nordic foods that are rich in plant-based and whole grain food items, fish and rapeseed oil, but with reduced amounts of meat, meat products, added sugar, salt and saturated fat. The nutrient composition in ND fulfills the Nordic nutrition recommendations [53]. In the OTIS study, we modified the ND to include a reduced overall intake of protein (total reduction 30%) in age-appropriate porridge and milk cereal drink (MCD), baby food in glass jars (BIG) and homemade baby food recipes during CF. Protein intake was still within the national recommended levels [29].

### 2.3. Nordic Portfolio in the NG

To enhance intake of the ND during CF, a portfolio of interventions (Figure 2) was offered to support the families with; (a) a 24-day taste portion schedule with repeated exposures to Nordic fruits and vegetables from 4 to 6 mo of age; (b) Nordic homemade baby food recipes for main course meal from 6 to 18 mo of age; (c) Nordic food family recipes and (d) Nordic protein-reduced baby food products from Semper AB, Sundbyberg, Sweden (Table A1).

Parents were also supported with a food map with inspiring images of 28 different Nordic fruits, berries and vegetables to place in the family’s home. For each image, parents were provided with homemade recipes of ND baby food purées to provide a variation of taste experiences during the CF period. When the infant had experienced a recipe, the parents checked off the respective box for these 28 fruits and vegetables. Nine of these recipes were a part of the taste portion schedule and requested to try during the period of repeated exposures. The rest of the recipes were optional to use during the study (Table A1). In the NG group, the parents were also invited to participate in a closed Facebook group. The intention with the Facebook group was to provide educational support and motivate and support the parents through videos, images and notifications to get familiar with preparation of the different food items, try out the provided Nordic study recipes and to increase the parents’ motivation to complete the taste portion schedule. The videos illustrated how to cook the baby food purées in the taste portion schedule, recorded by the first author (U.J.). The first author and the research nurses were responsible for the content in the Facebook group and replied to all questions.

### 2.4. Taste Portion Introduction with Repeated Exposure in the NG

When infants in the NG (*n* = 41) were between 4 and 6 mo of age, they were introduced to Nordic foods by a taste portion schedule (Figure 3). Parents were supplied with nine recipes for Nordic homemade fruit, berry and vegetable purées. They bought all the ingredients and prepared the purées themselves. Composition and preparation of the recipes have been described in detail elsewhere [54]. The taste portion schedule was developed to introduce sweeter flavors and then to increase sour and bitter flavors (Figure 3). Parents were given oral and written instructions how to offer and feed the infant the taste portions (5–15 mL purée per exposure) besides continued breast-feeding or formula-feeding.

### 2.5. Taste Portion Introduction and Guidelines in the CG

Parents to infants in the CG (*n* = 40) received a brochure from the Swedish Food Agency [55], which contained recommendations for taste portions and solid food introduction during CF (Figure 2). At the study start, infants were either exclusively breast-fed or formula-fed and had not been fed taste portions or other solid foods. Similar to NG, parents decided themselves when to start with the first taste portions between 4 and 6 mo of age. The Swedish recommendations recommend 6 mo of exclusive breast-feeding or formula if the child is not breast-fed. From the age of 4 mo at the earliest if the child became interested in other foods, parents could offer tiny taste portions besides breast-feeding or formula-feeding. The tiny taste portions were planned to be small amounts of family foods, “just a pinch” (1 mL) offered from a spoon or the parent’s fingers to the infant’s mouth, to slowly give the child the opportunity to try new flavors and textures. The recommended amount, “just a pinch” to a teaspoon per day can gradually be increased to a couple of teaspoons per day. From 6 mo of age, the guidelines suggest offering taste portions of vegetables, potatoes, rice and fruits beside breast-feeding or formula-feeding. The guideline provides parents with one “universal” recipe for homemade taste portions and to change to different vegetables in the universal recipe (carrot, parsnip, corn, green peas, broccoli, cauliflower or potatoes). The brochure also included three recipes for main course meals of homemade baby food (one fish-, one meat- and one vegetable dish) from 6 mo of age. In this group, no additional instructions were given from the research team, besides full access to online information to the national recommendations from the Swedish Food Agency website. However, both the NG and CG groups continued their interactions with their respective community child health center for vaccinations, growth monitoring and infant health and nutrition advice. These services are directed by national guidelines, are provided free of charge and was available to all study participants. Baby food products provided in the study were manufactured by Semper AB, Sundbyberg, Sweden (Table A1).

### 2.6. Foods Records and Diet Calculations

Five-day FR were recorded at 12 and 18 mo of age in order to assess dietary intake and including the intake of energy and nutrients. FR were also used to explore eating behaviors, particularly the difference in eating pattern between the two study groups. Parents reported everything their child ate and drank, including breast milk (BM) and any food supplements, for example, vitamins, using a standardized FR, within two weeks of the infant’s 12- and 18-mo birthday. Parents noted daily the meal type, time of day, which foods and drinks their infant was offered including amounts and brand names. BM intake was reported as ‘meals’ or ‘snacks’ estimated to 102 or 25 g of milk, respectively [56,57]. The food and drink intake were converted to grams using standardized weights for consumed foods from the Swedish Food Agency Database [58]. The food calculations for the mean daily energy intake (EI) and macronutrient sub-classes, the software Dietist Net Pro (Kost och Näringsdata AB, Bromma, Sweden) and the food composition database (version 02/05/2019) from the National Food Administration, Sweden were used. The food databank was supplemented with special products for infants used in the study with nutrient contents analyzed and supplied by Semper AB, Sundbyberg, Sweden. Data on fruit and vegetable content in the study products were supplied by Semper AB, Sundbyberg, Sweden. Contents in other baby food brands and products were assessed from these companies’ websites. Two pediatric dieticians calculated EI, macronutrients and fruit and vegetable content from the FR. Fruit juices, vegetable juices, potatoes, chili, garlic, ginger and herbs were not included in the intake from the fruits and vegetables.

### 2.7. Sensory Profiling of the Baby Food

In order to more objectively show differences in flavor and consistency between different fruits and vegetables used in the taste portion schedule and in the home exposure meal, we used a generic descriptive analysis [59,60] for sensory profiling. Nine homemade baby food purées, two commercial baby food purées and three dishes made of the homemade purées used in the home exposure meal were analyzed at the sensory laboratory (ISO 8589) at the University of Turku, Finland. Samples were prepared following the recipes from the Nordic taste portion schedule (Figure 3) and were mixed with infant formula (Baby Semp 1, Semper AB, Sundbyberg, Sweden). Two commercial baby food purées, corn-potato and mango (Semper AB, Sweden) were used as references in the sensory profiling. To these, infant formula was added to obtain comparable samples to the homemade purée samples. Samples were frozen in small cubes and thawed to +4 °C before use. The three dishes, lingonberry purée with milk, cranberry yogurt with yogurt and cauliflower purée with biscuit, were prepared following the recipes and instructions from the home exposure meal. Samples prepared for the sensory evaluation were served in a covered glass container with 40 g purée, except for the biscuit (2 g) with cauliflower purée (8 g). Vegetable purées were served similar to a baby food serving temperature at approximately 37 °C whereas fruit and berry purées were served at room temperature (~21 °C). Sensory profiling was done with 13 screened and pretrained adult assessors (all female, age 26–49, mean 37.5 years) [60,61]. Their ability to describe sensations verbally was tested as well as their normal function of senses. They were also trained in advance to use different sensory evaluation methods and to describe and evaluate different types of food samples. The training phase included generation of a consensus profile sensory lexicon for the baby food purées and determination of the evaluation techniques for each of the sensory attributes in the lexicon. In addition, reference samples for each attribute were determined and selected (Appendix A). The intensities of sensory attributes were rated on a line scale (from 0 = “none” to 10 = “very strong”) utilizing the anchored reference samples. Focus of the sensory profiling was on orosensory properties. Before the real sensory session, the panel had a practice session in sensory booths of the laboratory. Based on the training, a total of 17 sensory properties were selected for the profile. Orthonasal smell included intensity of aroma, sweet aroma, sour aroma and fresh aroma. Texture properties included graininess, watery, thickness, mouth filling and stickiness. Taste properties were sweetness, sourness, saltiness, bitterness and umami. Rated flavor properties were total intensity of the flavor, astringency and after-flavor. Quantitative evaluation protocol followed the procedure used before [60,61]. Data were collected with Compusense Cloud version 8.4 (Compusense Inc., Guelph, ON, Canada).

### 2.8. Videotaped Home Exposure Meals

To assess infants’ food acceptance, two videotaped exposure meal sessions took place in each family’s home (*n* = 81) between May 2016 to March 2019, within two weeks of the infant’s 12 and 18 mo birthday (Figure 2). Parents were instructed to not feed their child within 2 h before the meal session started. Infants were served an experimental Nordic flavored snacks meal either at 9–10 a.m. or 2–3 p.m. The parents decided which time was most suitable for the family, and also which parent was to participate in the videotaped meal. The participating parent was instructed to interact as he or she would normally do at an eating occasion at home and to end the meal when the parent perceived that the child was done. Parents were blinded to the hypotheses of this effort. The exposure meal was served in the family’s kitchen using their own baby chair and baby bib, and videotaped with an iPad Air 2 (Apple Inc., Cupertino, CA, USA) on a tripod on the kitchen table to visualize both parent and child from the front. The exposure meal was served on a standardized plate, bowl, cup and spoon and weighed (electronic scale IKEA Ordning 20047, IKEA, Älmhult, Sweden) immediately before and after eating including all spillage from the table, floor, bib, etc. Food for the meal sessions was prepared in two steps. First step; Nordic berries and vegetable purées were cooked and prepared to frozen cubes containing lingonberry purée, cranberry purée and cauliflower purée at the Department of Clinical sciences pediatric research unit at Umeå University. Recipes were developed for the study and were part of the taste portion schedule at 4–6 mo of age in the NG (Figure 3) and therefore the taste was familiar to infants in the NG. Second step; the frozen berries and vegetable cubes were thawed. Three grams of lingonberry purée was mixed with 30 g cow’s milk (3% fat content) to be served as a drink, 50 g plain yogurt (3% fat content) was mixed with 10 g cranberry purée to be served in a bowl with spoon and 2 g unflavored wheat biscuit with 8 g cauliflower purée to be served as a spread on a plate. This last step was prepared immediately before serving in the family’s kitchen by a pediatric dietician. During the videotaped meal session no one else was allowed in the kitchen besides the infant and one parent. At every home exposure meal session, the same pediatric dietician participated and was blinded to the group allocation. Testing procedures remained the same at both video sessions when the infant was 12 and 18 mo of age.

### 2.9. Videotape Analyses

The videotaped meals were coded and analyzed at Karolinska Institutet, Stockholm, Sweden by a trained pediatric psychologist (L.L.) and two Master students in psychology from Stockholm University, Stockholm. To determine infants’ food acceptance, a coding scheme was used for normal eating behavior of infants under normal conditions in the home [62]. We analyzed four different eating behavior codes that covered infant self-eating acceptance, infant accepting food fed from the parent, infant accepting drinking and infant refusing or rejecting food from the parent. Those four behavior codes were counted during 6 min divided into three separate time sessions; 2 min in the beginning, 2 min in the middle and the last 2 min of the meal session. Each videotaped meal session was approximately 10–15 min. Reliability was tested between the three independent observers coding behavior. The observers were blinded to group allocation. Reliability was tested between the three independent observers with intraclass correlation coefficients (ICC) varying from 0.92 to 0.99. Parents’ sensitivity for the infants’ signals were coded to control if sensitivity may affect infant’s food acceptance according to Ainsworth’s maternal sensitivity scale [63] with nine points ranging from very insensitive (1) to very sensitive (9). Components included in sensitivity are awareness of the infant’s signals, ability to have an adequate interpretation of the infant’s signals, react correctly to the situation and the infant’s needs and to respond without delay. The procedure to code sensitivity was the same as for infants’ food acceptance described above and inter-rater agreement was (IRA) 0.93.

### 2.10. Questionnaires and Demographic Variables

When the infant was 6 mo of age, parents answered an electronic questionnaire on their child’s temperament using the Baby Behavior Questionnaire (BBQ) [64]. The subscales and scores on manageability, sensory sensitivity and approach-withdrawal were analyzed together with the videotaped exposure meal sessions to study if temperament may contribute to food acceptance. To further include information on the participants’ eating behavior, the Children Eating Behavior Questionnaire (CEBQ) [65] was electronically answered by the parents when the infants were 12 and 18 mo of age. Demographic data were collected from both caregivers at baseline, which included information about family composition, education level, age of parents and country of birth. Data were electronically administered by an electronic survey (Textalk Websurvey, Mölndal, Sweden) [66].

### 2.11. Anthropometry, Blood Samples and Laboratory Analyses

Anthropometric data were collected within 2 weeks of the infants’ 12- and 18 mo birthdays at the pediatric research facility at Umeå University Hospital according to standardized procedures [67]: nude weight was measured to the nearest 5 g using electronic scales (Seca 727, Seca, Hamburg, Germany), head circumference measured to the nearest 0.1 cm using a non-stretchable measuring tape (Seca 212, Seca, Hamburg, Germany), recumbent length was measured to the nearest 0.1 cm using an infantometer (Seca 416, Seca, Hamburg, Germany) and BMI was calculated as weight in kg divided by (length in m)^2^. Venous blood samples were collected by experienced pediatric nurses after a 2-h fast at 12 mo and at 18 mo of age in an EDTA-containing tube and a serum separator tube. Blood samples were collected the same day as the anthropometrical measurements. If the infant was ill or had recently been immunized, sampling was postponed for 2 weeks to avoid the influence of an acute-phase response on blood indices. Serum-urea and s-folate were analyzed to validate different protein intake as well as fruit and vegetable intake between the two groups. Blood in the serum separator tube was centrifuged for 10 min at 2000 rpm, and s-urea and s-folate were analyzed within 4 h of collection at the Department of Clinical Chemistry, Umeå University, Sweden using a Roche Cobas 8000 (Roche Diagnostics) [68]. The laboratory reported s-folate values above 45 nmol/L as “>45 nmol/L”. In the analyses, these were set to 45 nmol/L.

### 2.12. Group Size Calculation

Sample size was calculated to find a significant difference with medium effect size (0.50) in food acceptance from the home exposure meals at 12 and 18 mo of age between NG and CG (power 80%, alpha = 0.05) [69]. To allow for an attrition rate of 5%, 40 participants were recruited per study group. In the NG group we recruited one extra (*n* = 41), because one participant had data missing from the BBQ.

### 2.13. Ethical Considerations

This research study was approved by the Regional Ethical Review Board at Umeå University (2014-363-31M), Umeå, Sweden. Written informed consent was obtained from both caregivers.

### 2.14. Statistical Analyses

SPSS 24.0 (SPSS, Chicago, IL, USA) was used for statistical analyses. Outcomes were analyzed and reported as means (± standard deviations, sd) or if non-parametric data as median (min-max and interquartile range, IQR), categorical data as numbers and percentages. EI and macronutrients sub-classes are reported as kilojoules (kJ) and grams (g) per day, respectively. Calculations of EI per kg body weight were presented as kilojoules (kJ) and protein (g) per kg body weight per day. For group comparisons, independent *t*-test (normally distributed data) and the Mann-Whitney (non-normally distributed data) was used. Chi^2^ test and Fisher’s test were used for comparisons between categorical variables. For comparisons within the NG for the taste portion schedule for type of foods, amounts per food item, refuses per food item and total exposures, ANOVA was used. In the statistical analysis of the main course meals groups, the independent Kruskal Wallis Samples T-test for non-parametric data was used. Pearson’s correlation was used to analyze the relationship between infants’ temperament and their food acceptance and refusal behavior, and correlation between parents’ sensitivity and infants’ food acceptance and refusal behavior. Reliability among the observers from the video analyzes was tested with intraclass correlation coefficients (ICC) and inter-rater agreement (IRA). To calculate effect size for parametric data Glass’s delta (Δ) was used, where $$\Delta =\frac{\left(Mean_{1}-\;Mean_{2\;}\right)}{s.d.control}$$. Glass’s delta [70] calculation was performed when it was more representative SD of a population (CG) comparing to the infants (NG) affected by an intervention. Sensory profiling data were analyzed with IBM SPSS Statistics 22.0 (IBM Corporation, Armonk, NY). One-way ANOVA (*p* < 0.05) together with Tukey’s HSD post-hoc test was used for each sensory attribute of the samples.

## 3. Results

### 3.1. Study Participants

Of the 81 infants who were randomly assigned, 90% finished the study until 18 mo of age (Figure 1). There were no differences between groups considering attrition rate, nor in demographic data, anthropometrical data or breast-feeding duration during the study period (Table 1). Mean s-urea was higher in the CG at 12 and 18 mo, but mean s-folate did not differ between groups at any age (Table 1). Neither infant temperament (BBQ) at baseline nor child eating behavior (CEBQ) were different between groups at 12 mo and 18 mo, except for one subscale for Enjoyment of Food, which was significantly higher in the CG at 18 mo (Table 2).

### 3.2. Effects of Taste Portion Schedule in the NG

All participants in the NG (*n* = 41) finished the taste portion schedule between 4.8 and 5.7 months of age. The taste portion schedule lasted 25.3 ± 4.2 days with total 62.3 ± 9.5 exposures, i.e., 87% of total scheduled exposures. The daily intake (mean) from the purées was 23.2 ± 12.8 g and parents reported 3.2 ± 4.2 refusals throughout the exposure period. Of the 41 participants, 31 (76%) were breast-fed during the exposure phase. Fluids added to the taste portion recipes were categorized into; BM 46% (*n* = 19), formula 32% (*n* = 13), mix of BM and formula 20% (*n* = 8), and oat cream 2% (*n* = 1). As shown in Table 3, there were no differences in acceptance, amounts consumed or recorded refusals between types of fruits or vegetables.

### 3.3. Parental Support of the Food Map and Social Media in the NG

In the NG, 98% (*n* = 40) of the families used the optional food map with Nordic baby food recipes of fruits and vegetables. The families prepared and fed the infant with 11 of 14 recipes from Nordic fruits and berries and 10 of 14 recipes from Nordic roots and vegetables during the study period. Ten families (25%) tested all the 28 baby food recipes in the food map. During the study period, 78% of the families joined the closed Facebook group, and 27% of the families had both parents participating in the group.

### 3.4. Sensory Profiling of the Baby Food

The sensory profiling showed differences across the sensory attributes from the taste portion schedule in bitter taste, sour taste, flavor intensity and astringent flavor (Figure 4a,b). Nordic berries and root vegetables were significantly different to the two commercial baby food products (mango, corn-potato). As demonstrated in Figure 4c–e there was significant changes across the attributes in the sensory profile from the home exposure meals when cow’s milk and yogurt was added to the lingonberry and cranberry purées. Particularly, the lingonberry milk and the cranberry yogurt had significantly less bitter taste (*p* < 0.001), flavor intensity (*p* < 0.001), astringent flavor (*p* < 0.001) and sour taste (*p* < 0.05).

### 3.5. Energy and Macronutrient Intake

There were no differences in daily mean intake of energy, fat or carbohydrate between the NG and CG groups at 12 and 18 mo of age, respectively (Table 4). Protein intake in both groups were within recommended levels although there were significant differences between groups. Protein intakes were lower in the NG compared to the CG at 12 mo and 18 mo, whether calculated as total intake g/day, as g/kg bodyweight or as energy percent (E%) and was higher at 18 mo in both groups (Table 4). The effect size was large at 12 mo (Glass’s delta 1.17) and at 18 mo (Glass’s delta 0.76), respectively (Table 4).

### 3.6. Fruit and Vegetable Intake

There were differences in daily fruit and vegetable intake between the groups at 12 and 18 mo of age (Figure 5a). NG infants consumed 54% more fruits and vegetables per day at 12 mo, and 32% more at 18 mo in comparison to the CG (Figure 5a). The effect size was large at 12 mo (Glass’s delta 1.11) and moderate at 18 mo (Glass’s delta 0.62). As shown in Figure 5b,c, even when categorized separately into fruits and vegetables, the differences remained between the groups at 12 mo and 18 mo, except for fruit intake at 18 mo. Particularly, the NG consumed 57% more fruits at 12 mo compared to the CG (Figure 5b). At 12 mo and 18 mo NG infants consumed 50% more vegetables compared to the CG (Figure 5c). The vegetable consumption, when categorized into root vegetables, was significantly higher in the NG at 12 mo and at 18 mo, respectively (Figure 5d). Moreover, when fruits were separated into different categories, the NG ate significantly more berries at 12 mo and at 18 mo compared to the CG (Table 5). The intake of exotic fruits was significantly higher in the CG at 12 mo and at 18 mo, respectively (Table 5).

As demonstrated over time in Figure 5, both groups started at 6 mo with no differences in intake of fruits and vegetables. During the study period both groups increased fruits and vegetables intake up to 12 mo where after both groups decreased their intake until 18 mo of age. However, there was a higher intake of fruits and vegetables among the infants in the NG at all ages after baseline. Despite the decreased intake among the NG between 12 to 18 mo, it remained higher in comparison to the CG at 18 mo (Figure 5a–d). Finally, no differences were found in intake of fruits and vegetables between genders within the groups during the study period.

#### Main Course Meals

Vegetable consumption derived mainly from the daily intake during the main course meals such as lunch and dinner. As shown in Table 6, there were differences in serving homemade cooked baby food dishes rich in vegetables in the NG in comparison to CG at 12 mo and 18 mo of age. Compared to the NG the CG infants consumed more of the family food at both 12 mo and 18 mo (Table 6). The dietary assessment showed that family food contained low amounts of vegetables in comparison to the homemade baby food dishes and industrial BIG or pouches. There were no differences between the groups in daily intake of industrially produced baby food products such as main course meals in BIG or pouches or when main course meals were replaced with industrial produced porridge, MCD or formula (Table 6).

### 3.7. Effects of Food Acceptance at Home Exposure Meals

There were no differences between the groups in food acceptance, refusal behavior, consumed food or the serving time at 12 and 18 mo of age (Table 7). At the exposure meal at 12 mo of age with all three components taken together, infants in the NG and CG consumed in total 62.6% and 56.2%, respectively, of the amount presented. At 18 mo both groups consumed less food from the exposure meal; the NG ate in total 51.4% of the meal and the CG 48.0%. None of the three coded behaviors included in the term food acceptance showed no between group difference at 12 or 18 mo of age (Table 7). We found no significant associations between infant temperament and food acceptance or refusal behavior within or between the groups. Parents’ sensitivity scores were not different between the groups at 12 mo and 18 mo (Table 7). There were no group differences in the relationship between parents’ sensitivity and infant’s food acceptance or refusal behavior.

## 4. Discussion

This is the first multicomponent randomized trial, performed over time (>12 mo) with education and support for parents, to demonstrate the effects of early, repeated exposures to Nordic fruits and vegetables followed by a Nordic protein-reduced complementary diet compared to a diet based on current Swedish dietary recommendations on eating behavior and food acceptance. The most important finding is that the intervention affected infant eating behavior by increasing the intakes of fruits and vegetables and decreasing protein intake. This was also validated by the biomarker s-urea. The experimental ND, especially the introductory part, had flavors and textures that were significantly different from foods more commonly used as complementary foods in Sweden. Despite the differences in taste and flavor of the two diets, the overall adherence to the ND was high, accurately measured with dietary registrations and biomarkers, showing higher intake of fruits and vegetables in the experimental group throughout the study period. Thus, the infants in the NG willingly accepted the unfamiliar flavors of the taste portion schedule, which were far from sweet tastes that dominate contemporary complementary food. This high acceptance of bitter, sour and astringent flavors put the use of Nordic foods in the forefront during the sensitive period of infancy when food preferences are established. These results are in line with efforts by the WHO to reduce non-communicable diseases through improved fruit and vegetable intakes among children and adults [17,24,45]. In this study, we also present the first experimental evidence to support ND during CF as an environmentally sustainable food choice [18,37]. ND with mostly plant-based foods, less red meat, replaced by plant-based protein sources or fish is much lower in carbon footprint compared to existing dietary patterns in middle and high-income countries [37]. The reduction of Green Gas Emissions (GHGs) from food consumption can be reduced up to 35% from healthy plant-based diets like ND, Mediterranean diet or lacto-vegetarian diets [37,71]. From this perspective, the ND is in agreement with the EAT-Lancet report, which supports a healthy plant-based flexitarian diet without eliminating protein sources from dairy products or meat, optimized for health and environmental sustainability [18].

In a study from the Netherlands, Barends et al. demonstrated in a randomized intervention that repeated exposures to vegetables from early life until 23 mo of age increased the intake and fondness of these foods [72]. The intake of vegetables increased significantly (by 38%) up to 12 mo in the experimental group, but from 12 to 23 mo of age the experimental group decreased the vegetable intake to the same level as the control group [72]. In the present study, vegetable intake was 50% higher in the NG at 12 and 18 mo compared to the CG. We have, however, no follow-up data at 23–24 mo of age when pickiness and food neophobia may become more prevalent [73]. We also found that with increasing age the total intake of fruits and vegetables decreased from 12 to 18 mo independent of EI. During this age, many children start preschool, which may also change food intakes from fruits and vegetables. In a European study from three countries, Caton et al. demonstrated that age was a significant predictor of eating behavior of vegetables among children in ages 4 to 38 mo [74]. Particularly, younger children were less fussy, enjoyed food more and had increased acceptance for novel food from vegetables [74]. This was also found in a randomized intervention study from UK, in which younger children aged less than 23 mo compared to children older than 24 mo showed a significantly higher intake of all types of vegetables [19]. The amount of pickiness and food neophobia at introduction can be one explanation for the change in eating behavior and acceptance of vegetables at different ages [73]. We found no differences in food fussiness measured with the CEBQ between 12 to 18 mo, either within or between the groups. Overall, the CEBQ showed no difference between groups, except for enjoyment of food at 18 mo, which was higher in the CG (*p* < 0.013). However, the score for enjoyment of food in the NG was similar or higher compared to a Swedish study on children of the same age [75].

When it comes to sensory experiences from food, early life plays an essential role to develop food preferences [76]. Repeated exposures with bitter taste from vegetables to inexperienced infants have been shown to form acceptance to these flavors during CF [77] and to increase vegetable intake [19,20,21]. The present study confirms this but adds that experiences of a variety of flavors from vegetables and fruits can be accepted after having been introduced during a relatively short period of time. Our results are in line with other studies where 8–10 repeated exposures to vegetables or fruits in early infancy are appropriate to establish food preferences [3], and we also endorse previous findings that a variety of flavors from vegetables increases acceptance of novel foods [78,79]. However, we contribute new knowledge that these findings remain during a long-term, home-based multicomponent intervention together with environmental sustainable eating during CF.

Although the present effort was not powered to detect minor differences in growth, the anthropometric measurements show no significant differences between the study groups. There were biochemical differences between the groups with lower s-urea concentrations in the NG at 12 mo and 18 mo, due to their lower protein intake. Serum-folate, a proxy for fruit and vegetable intake showed no difference, possibly due to a combination of a limited sample size and the ceiling effect of the laboratory analysis, where values above 45 nmol/L were not accurately reported.

Food acceptance was assessed using two videotaped exposure meals at 12 and 18 mo of age administered by a pediatric dietician at the participant’s home. The meals, served as a mid-morning or mid-afternoon snack consisted of a lingonberry milk drink, a cranberry yogurt and a biscuit with cauliflower purée. The intended design of the exposure meal was to provide participants in both groups with flavors that the NG would have more likely encountered during the introduction of CF than the CG, and then to see if these early experiences would translate into higher acceptance in the NG during the exposure meal. This was, however, not the case and the participants in both groups consumed similar amounts of the exposure meal and showed similar levels of accepting and refusing behavior. A likely explanation for this is the changes in flavor and consistencies brought on by the preparation of the exposure meal. The sensory profiling showed that preparation of both the cranberry yogurt and the lingonberry milk significantly decreased the bitterness, sourness and astringency of cranberry and lingonberry that the participants in the NG would have encountered during the introduction of CF, rendering them palatable also for the participants in the CG. In a randomized intervention study from the UK, Hetherington et al. showed how the sensory experience of vegetable purée was changed when mixing with infant milk formula to less bitter and sour tastes and less intense flavors [20]. In contrast to our study, this was in the early stage of CF and not in the later outcome of the study. However, it demonstrated that the sensory properties of vegetable purées changed when they were mixed with milk, similar to lingonberry milk and cranberry yogurt in the present study. Hetherington et al. also reported how the sensory experiences affected the later stage of the study, when both groups were exposed to a new vegetable (parsnip) [20]. They found no significant differences in the intake between the groups, explained by the fact that parsnip purée had low taste intensity, was sweet, not sour or metallic and was the least bitter of all the study vegetable purées (green beans and spinach). Similar to our findings, the experimental exposure meal failed to demonstrate differences in food acceptance, because the sensory properties were different from introduction to the exposure meal.

### Strengths and Weaknesses

The randomized design allowed us to evaluate any differences between the intervention and control group as an effect of the intervention per se. The adherence to the study protocol was high in both groups with few drop-outs despite a duration of the intervention of more than 12 mo from recruitment to follow-up, showing that this type of intervention is feasible and sustainable to implement during an extended and important time during which the participants undergo fast and extensive growth and development. We applied several educational efforts to the parents in the NG regarding baby foods and their preparation. Apart from how to plan, purchase and prepare ND baby and family foods with written recipes and videotaped instructions on how to follow these recipes, we also provided the parents in the NG with a food map, which they could follow to increase exposure to Nordic fruits and vegetables and access a group on Facebook, both intended to increase adherence. Given the high adherence to the food map (98% of participating families) and high participation in the Facebook group (78% participating families), we believe that these efforts positively affected adherence to the NG diet, although we cannot say to what extent or which components of this parental support were most crucial to the response. In a European randomized study, Fildes et al. reported how parental support during repeated exposure increased vegetable acceptance among infants in the UK compared to the country-specific recommendations for infants [23]. Parents in the intervention group were informed how to get used to unfamiliar vegetables, which increased the liking of a variety of vegetables among the infants. Similar to our findings, repeated exposure together with parental guidance increased vegetable intake in comparison to the group given national dietary guidelines for infants [23]. Compared to the present study, their study was a single-component intervention with short-time impact and did not include how to prepare homemade baby food purées.

At the home exposure meal, the pediatric dietician, the parent administering the meal, and the psychologists who analyzed the video-recordings were blinded, ensuring little bias. We believe that the home-based design was beneficial to implementation of the study in the sense that compared to testing in a laboratory or other research facility, the participants were familiar to the surroundings, with for example, the same plate, cup and cutlery as the child was familiar with, the parents decided themselves which one would be present and at which time the exposure meal would be served. All these efforts made the setting as optimal as possible for the exposure meal. Then the same pediatric dietician prepared the same exposure meals to all the participants, but with the parent of choice serving the meals. The analysis of the video-recordings showed high scores on reliability measured as intraclass coefficients (0.92–0.99) and inter-rater agreement (0.93), but we also find associations between parental sensitivity to their child’s signals and the child’s food acceptance and refusal behavior during the meal within the groups (data not shown), which have been reported in other studies [47,80]. However, as shown by the sensory properties of the exposure meal, the preparation probably rendered the meal insufficiently challenging, which in turn made us underestimate effect size the test could show and thus the sample size. Another weakness of the study is the difficulty of delineating the effects of the individual interventions on the outcome. From the available data we cannot sufficiently assess the relative contribution of, for example, the parental educational efforts compared to the recipes we provided to the NG on the overall intake of fruits and vegetables.

The practical implication based on this research is that it is possible to introduce and uphold infants to a complementary diet with sensory experiences that are way beyond what is presently used among Swedish infants. Such a diet would be healthier in terms of food composition but also more environmentally sustainable as it uses regionally grown crops and plant-based food. The CG followed the national dietary guidelines without parental education or Nordic homemade baby food recipes, and, consequently, ate significantly more of imported exotic fruits such as bananas, mango, etc., whereas the NG had virtually no daily intake of imported fruits per day. The level of support used in the present study is beyond what is feasible within most child health care settings and more studies are needed to show which levels of intervention and support will achieve maximum cost-effectiveness. Finally, this study shows the feasibility of performing multicomponent interventions in the home setting, which we believe is necessary in order to bridge the gap in knowledge of the most optimal ways to transfer the child from the liquid food of infants to the family’s food [46] in order to understand the multiple factors that are involved in establishing longstanding healthy eating behavior among children.

## 5. Conclusions

In conclusion, providing a ND to infants is feasible and results in a significantly higher intake of Nordic fruits, berries, vegetables and root vegetables compared to infants recommended the current Swedish complementary diet. However, when testing food acceptance in infants randomized to an intervention with large differences in taste and texture one must be careful not to change the sensory properties of the food items tested so that the possibility to show differences are not eliminated. In addition, this study also shows that education and support to parents may impact on their child’s food intake, even among these highly educated participants. The high acceptance of the bitter, sour and astringent flavors in Nordic fruits, berries and vegetables make them useful in the introduction of CF during the sensitive period of infancy when food preferences are established.

## Figures and Tables

**Figure 1 foods-10-00275-f001:**
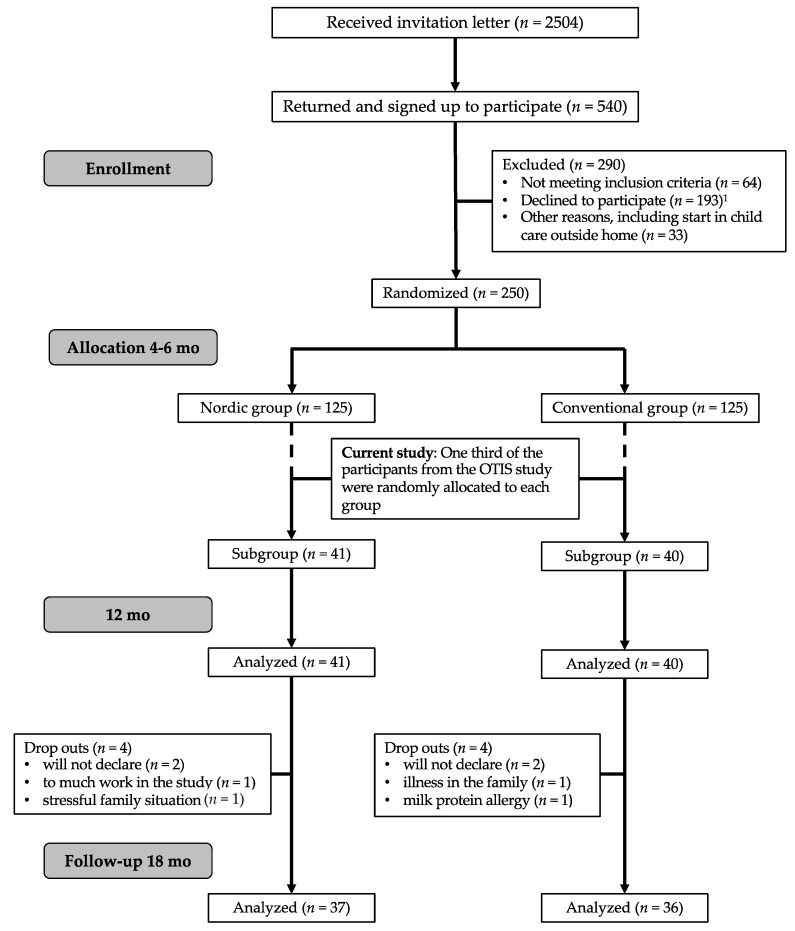
Flowchart of the current study in relation to the optimized complementary feeding study (OTIS) study. ^1^ Declined to participate; too much work in the study or too long study period or will not give their child the specific study diet.

**Figure 2 foods-10-00275-f002:**
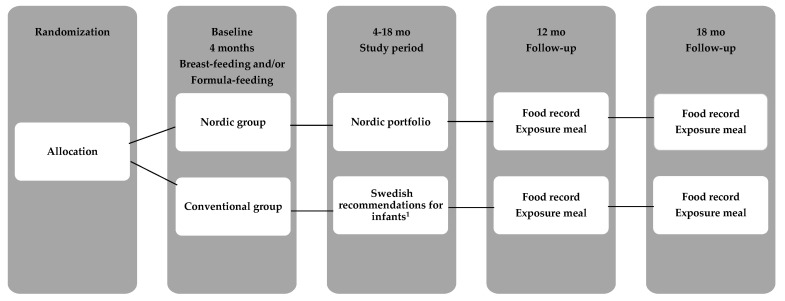
Flow diagram of the study. ^1^ The Conventional group (CG) parents received written information from the Swedish Food Agency with the current recommendations for infants.

**Figure 3 foods-10-00275-f003:**
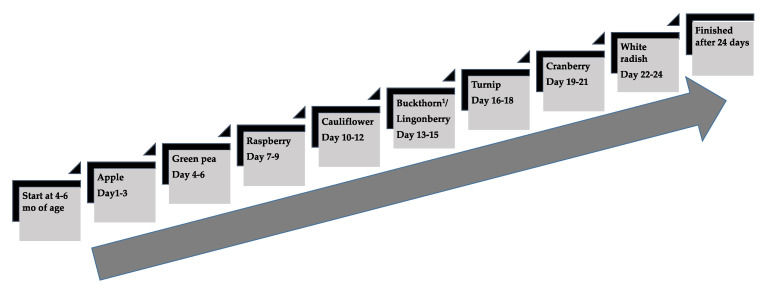
The taste portion schedule with the repeated exposures of fruits and vegetables in the Nordic group from 4 to 6 mo of age. Three exposures per day during three consecutive days per each fruit/berry or vegetable purée, in total 72 exposures during 24 days. ^1^ Buckthorn berry, a seasonal product was replaced by lingonberry when shortage appeared.

**Figure 4 foods-10-00275-f004:**
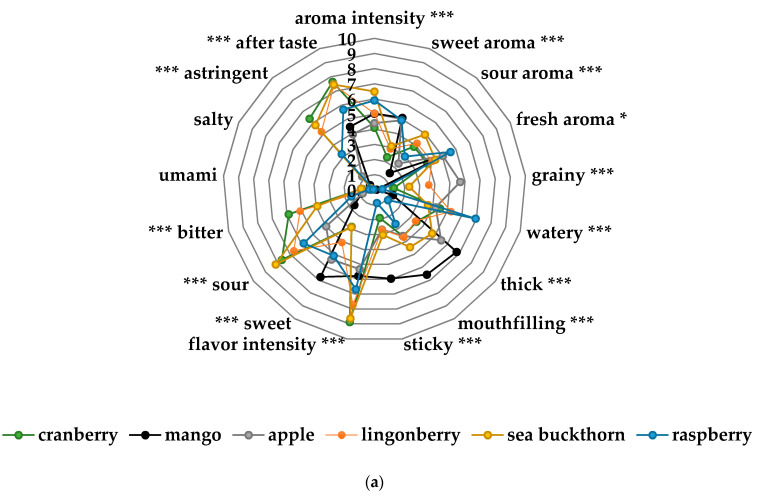
(**a**) Sensory profiles of the different fruit and berry purées in the taste portion schedule together with mango purée, a commercial product from Semper AB; (**b**) sensory profiles of the different vegetable purées in the taste portion schedule together with corn-potato purée, a commercial product from Semper AB; (**c**) sensory profile of the cauliflower spread of the exposure meal compared to the cauliflower purée in the taste portion schedule; (**d**) sensory profile of the cranberry yogurt in the exposure meal compared to the cranberry purée in the taste portion schedule; (**e**) sensory profile of the lingonberry milk of the exposure meal compared to the lingonberry purée in the taste portion schedule. Significant differences across the sensory attributes * *p* < 0.05, ** 0.01< *p* < 0.05, *** *p* < 0.001.

**Figure 5 foods-10-00275-f005:**
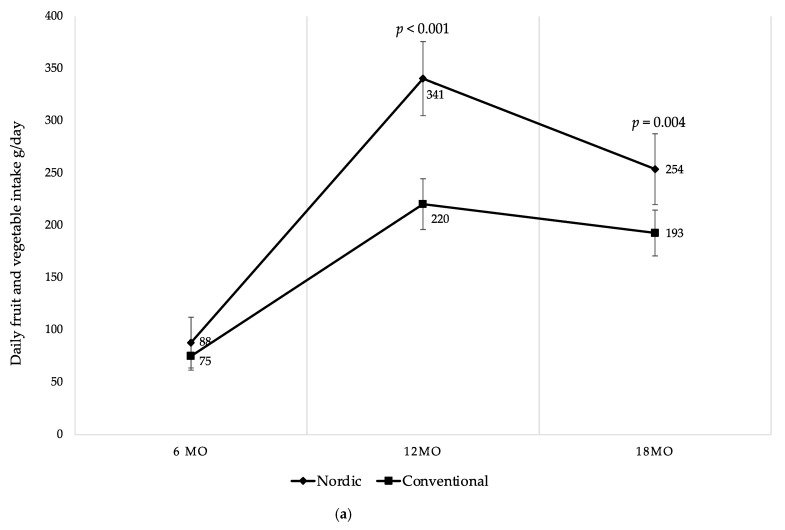
(**a**) Mean daily fruit and vegetable intake (g/day) at 6, 12 and 18 mo of age; (**b**) mean daily fruit intake (g/day) at 6, 12 and 18 mo of age; (**c**) mean daily vegetable intake (g/day) at 6, 12 and 18 mo of age; (**d**) mean daily root vegetable intake (g/day) at 6, 12 and 18 mo of age. Fruit juices, vegetable juices, potatoes, chili, garlic, ginger and herbs are not included in the calculations. Whiskers indicate 95% confidence intervals. *p*-values indicate between group differences.

**Table 1 foods-10-00275-t001:** Study participants’ characteristics, anthropometrical and biochemical data.

	Nordic Group (*n* = 41) Mean ± sd	Conventional Group (*n* = 40) Mean ± sd
Age at study start (mo)	4.1 ± 0.6	4.0 ± 0.6
Girls/boys (*n*, %)	16/25 (39)/(61)	19/21 (48)/(52)
Neonatal characteristics		
Birth weight (kg)	3.7 ± 0.5	3.7 ± 0.4
Birth length (cm)	51.0 ± 2.1	50.7 ± 2.0
Birth head circumference (cm)	35.7 ± 1.2	35.5 ± 1.4
Gestational age (weeks)	39.9 ± 1.2	40.0 ± 1.2
Breast-feeding duration (*n*, %)		
4 mo	41 (100)	40 (100)
6 mo	29 (71)	31 (78)
9 mo	15 (37)	18 (45)
12 mo	6 (15)	5 (13)
18 mo	1 (2)	0 (0)
Duration of exclusive breast-feeding (mo)	3.9 ± 1.4	4. 4 ± 1.3
Started at Day Care Centre <18 mo, (*n*, %)	15 (44)	19 (50)
Family characteristics		
No siblings (*n*, %)	19 (46)	21 (53)
Mothers age (year)	31 ± 3.9	32 ± 5.4
Partners age (year)	33 ± 4.8	34 ± 4.3
Education level Mother (*n*, %)		
Elementary school	0 (0)	1 (3)
High school	14 (34)	10 (25)
University	27 (66)	29 (72)
Education level Partner (*n*, %)		
Elementary school	5 (12)	2 (5)
High school	11 (27)	12 (30)
University	25 (61)	26 (65)
Ethnicity (born in Sweden) (*n*, %)		
Mother	41 (100)	37 (95)
Partner	36 (88)	33 (83)
Anthropometry		
12 mo		
Age (mo)	11.7 ± 0.3	11.8 ± 0.3
Body weight (kg)	10.1 ± 1.1	10.1 ± 1.2
Body length (cm)	76.2 ± 2.7	76.2 ± 2.9
Head circumstance (cm)	47.0 ± 1.1	46.8 ± 1.4
BMI (kg/m^2^)	17.4 ± 1.2	17.5 ± 1.3
18 mo		
Age at follow-up (mo)	17.8 ± 0.3	17.9 ± 0.3
Body weight (kg)	11.7 ± 1.2	11.7 ± 1.4
Body length (cm)	82.9 ±2.8	82.7 ± 2.9
Head circumstance (cm)	48.5 ± 0.9	48.4 ± 1.5
BMI (kg/m^2^)	17.0 ± 1.2	17.1 ± 1.5
Laboratory markers		
12 mo		
S-Urea (mmol/L)	2.5 ± 0.9 ^1^	4.2 ± 1.2 ^1^
S-Folate (nmol/L)	39 (35–45) ^2^	37 (31–43) ^2^
18 mo		
S-Urea (mmol/L)	4.1 ± 1.1 ^1^	5.2 ± 1.3 ^1^
S-Folate (nmol/L	34 (29–39) ^2^	31 (27–34) ^2^

Values are mean ± sd unless otherwise indicated. ^1^
*p* < 0.001, ^2^ median (IQR). BMI: body mass index.

**Table 2 foods-10-00275-t002:** Baby behavior questionnaire (BBQ) and children eating behavior questionnaire (CEBQ) data.

	Nordic Group (*n* = 41) Mean ± sd	Conventional Group (*n* = 40) Mean ± sd
Baby Behavior (baseline)		
Manageability	3.6 ± 0.5	3.5 ± 0.6
Sensory Sensitivity	3.3 ± 0.9	3.2 ± 0.7
Approach Withdrawal	4.2 ± 0.6	4.2 ± 0.6
Children Eating Behavior		
12 mo		
Satiety responsiveness	2.7 ± 04	2.9 ± 0.5
Slowness in eating	2.6 ± 0.4	2.7 ± 0.4
Fussiness	2.9 ± 0.3	2.9 ± 0.3
Food responsiveness	2.3 ± 0.8	2.3 ± 0.7
Enjoyment of food	4.0 ± 0.6	4.1 ± 0.6
Drink desire	1.9 ± 0.8	2.0 ± 0.7
Emotional undereating	3.3 ± 1.0	3.2 ± 0.9
Emotional overeating	1.6 ± 0.4	1.7 ± 0.6
18 mo		
Satiety responsiveness	3.0 ± 0.3	3.0 ± 0.4
Slowness in eating	2.6 ± 0.4	2.7 ± 0.5
Fussiness	2.9 ± 0.3	2.9 ± 0.3
Food responsiveness	2.2 ± 0.7	2.3 ± 0.8
Enjoyment of food	3.8 ± 0.6 ^1^	4.2 ± 0.5 ^1^
Drink desire	2.3 ± 0.9	2.1 ± 0.7
Emotional undereating	3.5 ± 0.8	3.3 ± 1.0
Emotional overeating	1.7 ± 0.6	1.7 ± 0.5

^1^*p* = 0.013.

**Table 3 foods-10-00275-t003:** Taste portion schedule in the Nordic group (*n* = 41).

Taste	Total Exposures (*n*) Mean ± sd	Total Amounts of Purée (g) Mean ± sd	Amounts of Purée Per Exposure (g) Mean ± sd	Total Refuses (*n*) Mean ± sd
Apple	8.2 ± 1.4	73.8 ± 47.0	8.8 ± 5.1	0.2 ± 0.7
Green peas	8.0 ± 1.2	73.4 ± 54.8	8.9 ± 6.0	0.4 ± 0.8
Raspberry	7.8 ± 1.6	70.0 ± 47.0	8.8 ± 5.3	0.3 ± 0.6
Cauliflower	7.8 ± 1.5	81.1 ± 48.0	10.1 ± 5.2	0.3 ± 0.8
Buckthorn/Lingonberry	7.7 ± 1.6	65.4 ± 46.2	8.3 ± 5.2	0.3 ± 1.0
Turnip	7.7 ± 1.4	72.9 ± 40.9	9.2 ± 4.4	0.4 ± 0.8
Cranberry	7.5 ± 1.5	61.5 ± 38.7	8.1 ± 4.8	0.8 ± 1.9
White radish	7.6 ± 1.7	69.9 ± 45.3	9.0 ± 5.0	0.5 ± 1.2

**Table 4 foods-10-00275-t004:** Daily mean intake of energy and macronutrients at the 12 and 18 mo 5-day dietary registrations.

	Nordic Group Mean ± sd	Conventional Group Mean ± sd	*p* for Difference ^1^
12 mo	(*n* = 37)	(*n* = 39)	
Age at food recording (mo)	11.6 ± 0.3	11.6 ± 0.3	0.50
Energy (kJ)	3900 ± 625	3920 ± 561	0.88
Energy/bodyweight (kJ/kg)	384 ± 54	388 ± 59	0.77
Protein/bodyweight (g/kg)	1.9 ± 0.4	2.9 ± 0.6	<0.001
Protein (g)	19.6 ± 8.1	29.1 ± 6.2	<0.001
Protein (E%)	8.5 ± 1.3	12.5 ± 1.5	<0.001
Fat (g)	35.5 ± 7.2	34.7 ± 7.0	0.65
Fat (E%)	33.7 ± 4.9	32.7 ± 3.7	0.30
Carbohydrate (g)	127.7 ± 25.6	122.5 ± 16.9	0.30
Carbohydrate (E%)	55.6 ± 5.1	53.3 ± 4.1	0.34
Days (*n*) between food recording and laboratory markers	7.7 ± 3.7	6.4 ± 5.9	0.24
Days (*n*) between food recording and home exposure meal	12.6 ± 8.5	13.9 ± 9.1	0.53
18 mo	(*n* = 34)	(*n* = 38)	
Age at food recording (mo)	17.5 ± 0.3	17.6 ± 0.3	0.34
Energy (kJ)	3938 ± 552	4214 ± 762	0.81
Energy/bodyweight (kJ/kg)	338 ± 53	361 ± 68	0.11
Protein/bodyweight (g/kg)	2.5 ± 0.8	3.1 ± 0.6	0.001
Protein (g)	29.1 ± 9.2	36.1 ± 7.4	0.001
Protein (E%)	12.5 ± 3.3	14.6 ± 1.2	0.002
Fat (g)	35.9 ± 7.3	37.5 ± 8.2	0.39
Fat (E%)	33.6 ± 4.6	32.9 ± 3.8	0.49
Carbohydrate (g)	118.8 ± 21.1	124.8 ± 24.2	0.27
Carbohydrate (E%)	51.5 ± 5.8	50.4 ± 3.9	0.35
Days (*n*) between food recording and laboratory markers	8.4 ± 5.7	10.0 ± 9.1	0.36
Days (*n*) between food recording and home exposure meal	13.2 ± 8.5	10.8 ± 8.0	0.22

^1^ Independent samples *t*-test.

**Table 5 foods-10-00275-t005:** Daily intake of berries and exotic fruits at the 12 and 18 mo 5-day dietary registrations.

	Nordic Group Median (Min-Max)	Conventional Group Median (Min-Max)	*p* for Difference ^1^
Berries (g/day) ^2,3^	*n* = 37	*n* = 39	
12 mo	12 (0–52)	1 (0–26)	<0.001
18 mo	9 (0–63)	3 (0–20)	0.004
Exotic fruits (g/day) ^3^	*n* = 34	*n* = 38	
12 mo	0 (0–19)	94 (4–251)	<0.001
18 mo	0 (0–41)	92 (19–175)	<0.001

^1^ Mann–Whitney test. ^2^ Strawberry are not included in the category. ^3^ Fruit juices are not included in the assessment.

**Table 6 foods-10-00275-t006:** Daily type of main course meals at the 12 and 18 mo 5-day dietary registrations.

	Nordic Group ^1^ Median (Min-Max)	Conventional Group ^1^ Median (Min-Max)	*p* for Difference ^2^
12 mo	*n* = 37	*n* = 39	
Porridge, MCD ^3^ or formula	0 (0–0.8)	0 (0–0.8)	0.60
Family food (main course meal)	0 (0–1.0)	0.4 (0–2.0)	<0.001
Commercial baby food	1.8 (0.6–2.0)	1.6 (0–2.0)	0.076
Homemade baby food recipes	0 (0–1.2)	0 (0–0)	<0.001
18 mo	*n* = 34	*n* = 38	
Porridge, MCD ^3^ or formula	0 (0–0.8)	0 (0–0.8)	0.98
Family food (main course meal)	1.0 (0–2.0)	1.4 (0–2.0)	0.026
Commercial baby food	0.9 (0–2.0)	0.4 (0–2.0)	0.076
Homemade baby food recipes	0 (0–0.8)	0 (0–0)	0.015

^1^ Two main course meals were eaten per day. ^2^ Kruskal–Wallis test. ^3^ Milk cereal drink.

**Table 7 foods-10-00275-t007:** Results from the home exposure meal at 12 and 18 mo of age.

	Nordic Group (*n* = 41) ^a^, (*n* = 37) ^b^	Conventional Group (*n* = 40) ^a^, (*n* = 36) ^b^	*p* for Difference ^1,2,3^
Parent’s sensitivity score			
12 mo (infant age)	5.2 ± 1.5	4.9 ± 1.6	0.32 ^1^
18 mo (infant age)	5.5 ± 1.7	5.4 ± 1.5	0.80 ^1^
Mother/Father (*n*,%) (participated at the exposure meal)			
12 mo	23/18 (56/44)	23/17 (57/43)	1.00 ^2^
18 mo	25/12 (68/32)	26/10 (72/28)	0.80 ^2^
Serving time 9–10 a.m./2–3 p.m. (*n*,%)			
12 mo	14/27 (34/66)	15/25 (37/63)	0.82 ^2^
18 mo	8/29 (22/78)	11/25 (31/69)	0.43 ^2^
Consumed food ^4^ (exposure meal)			
12 mo			
Cranberry yogurt (g)	39.4 ± 20.4	37.2 ± 20.2	0.62 ^1^
Biscuit with cauliflower purée (g)	7.1 ± 3.1	7.6 ± 2.7	0.47 ^1^
Lingonberry milk (g)	18.0 ± 11.8	13.1 ± 10.7	0.057 ^1^
18 mo			
Cranberry yogurt (g)	33.3 ± 21.6	29.7 ± 22.0	0.49 ^1^
Biscuit with cauliflower purée (g)	3.6 ± 3.6	4.7 ± 3.7	0.24 ^1^
Lingonberry milk (g)	16.0 ± 12.9	15.0 ± 12.0	0.72 ^1^
Food acceptance ^5^			
12 mo	23.5 ± 9.2	23.4 ± 9.0	0.96 ^1^
18 mo	20.2 ± 12.6	19.6 ± 8.7	0.82 ^1^
Refusal behavior ^6^ median (min-max)			
12 mo	2 (0–18)	5 (0–18)	0.49 ^3^
18 mo	2 (0–22)	3 (0–19)	0.43 ^3^

Values are mean ± sd unless otherwise indicated. ^a^ Number of participants at 12 mo and ^b^ 18 mo of age. ^1^ Independent samples *t*-test, ^2^ Fisher’s exact test, ^3^ Mann–Whitney test. ^4^ The exposure meal contained 60 g cranberry yogurt, 10 g biscuit with cauliflower purée (spread) and 33 g lingonberry milk per portion, totally 103 g food and drink. ^5^ Food acceptance includes infants self-eating acceptance, infant accepting food fed from parent and infant accepting drinking. ^6^ Infant refusing or rejecting food from parent.

## Data Availability

Data described in the manuscript will be made available upon request pending application and approval.

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
