# Peer review of "Acceptance of a Nordic, Protein-Reduced Diet for Young Children during Complementary Feeding—A Randomized Controlled Trial"

_foods, 2021, doi:10.3390/foods10020275_

Round 1

Reviewer 1 Report

This study randomly allocated infants (and their parents) to a diet high in fruits and vegetables compared to a group following the Swedish recommended intake guidelines for infants. Participants were tracked over 12 months. A series of measures investigated acceptability of fruits and vegetables and differences in the typical diets of infants at 12 and 18 months of age.

This is a very impressive study and is well-written. In some places the authors I feel need to explain the purpose behind their decisions a little more as readers are not as familiar with the study. The biggest potential flaw I see (and am very interested to see the authors response to this, the issue is also outlined below) is the level of support parents in the intervention group had compared to the control group, and the extent to which this may have impacted on study findings.

Overall, I think this work will make an important contribution to the scientific literature in this area.

INTRODUCTION

Suggest providing a definition for ‘complementary feeding’ as this term may not be universally known amongst the readership.

MATERIALS AND METHODS

Understand participants cannot be completely blinded to the group allocation, but they could have (potentially) been blinded to the purpose of the study, or even the fact that there was a comparison group? While this may be accompanied by ethical considerations. Is this the reason why participants were not entirely blinded to the purpose of the study (or maybe they were)?

“protein intake was still within the national recommended levels.” So really, this study is not a comparison between the ND and a low protein ND diet? What sources of protein were NG participants receiving so they still met recommended levels? Was it protein through sources other than animal protein?

It seems like the researchers went to great lengths to support parents on the NG diet. Were similar avenues of support setup for the control group? Otherwise, it may not be surprising that parents were pushing the NG diet more if they received greater support? It seems lots of researcher support for the NG but the CG was more or less left to their own devices? A Taste portion schedule was setup for the NG but not the CG?

2.6 Foods records and dietary assessment

When suggesting parents were asked to record everything their child ate and drank, was a uniform or standardised data collection tool provided, or did they just record on whatever was to hand?

2.7 Sensory profiling of the baby food. It isn’t clear why this was done. For what purpose was the process described in this paragraph undertaken?

Videotaped home exposure – was the exposure compared or controlled around what else the infant had eaten recently? A child is more likely to consume food if they are hungry.

Videotaped analyses – So the four different eating behaviour codes – did the infant score 1 point every time the researchers noted one of these instances, and then they were summed at the end of the video? How was data quantified here? Further, for the parents sensitivity analysis, how was this coded. Was this an overall rating the parent received from 1-9 for the entire exposure, and this was made at the judgement of those watching the video? For both, what training did those watching the vidoes receive to be equipped to make these judgements? Also what standardised tools were utilised?

RESULTS

It seems like the sensory profile information is not really relevant to this study? It is unclear how it ties in? It seems like its own separate study looking at the sensory profiles of different fruits and berries. How does this information relate to the purpose of this study?

I note protein intakes were lower in the NG compared to CG which we note is a positive. However, too little protein (below recommended levels) is problematic. Are we able to compare the protein levels to minimum recommended levels?

DISCUSSION

Understand the value of more fruit and vegetables and less protein, but some protein is still required to make an optimal healthy diet. Is there a risk that the ND could lead to too little protein being consumed?

STENGTHS AND LIMITATIONS

I may have missed it, but I’m not sure any data was presented about parents adherence to study protocols? Was there any missing data from the 5-day food diaries?

Again, the authors make clear the efforts they went to to promote the ND to the NG. My concern is this group received a lot of support making it more likely they would work to implement this diet, as opposed to the other group that received no support to implement a diet that met guidelines. Can the authors explain the extent to which they feel this may have impacted on study results when comparing between the two groups?

“We are convinced that the analysis of the video-recordings is valid…” Suggest removing this subjective statement instead allowing the reader to judge this themselves based on the information provided.

OTHER COMMENTS

Upon finishing the manuscript, I think it would have been of great value either early in the paper or alongside when each of the measures were introduced to make clear the purpose of these measures. Either, the specific research questions could be outlined at the end of the introduction, and then made clear which measures align to what research questions when they are presented, or it could be made clear upfront at the introduction of each measure. As I was reading through the different measures, for many it was not entirely intuitive the purpose of their inclusion, and what components of the greater research they were addressing. Putting research questions (and sub questions) upfront would help alleviate this.

Author Response

To Reviewer: Please see the attachment.

Reviewer 2 Report

Dear Authors,

The work entitled "Acceptance of a Nordic, protein-reduced diet for young children during complementary feeding a randomized controlled trial" is really well written and, despite its complexity, is clear in its exposition and well explained. Results obtained by the authors are interesting and useful to define some aspects such as the eating behavior and food acceptance of infants up to 18 months of age.

I appreciated the study that questions the current dietary recommendations provided by Swedish institutions for nutrition early age and I believe the goal of achieving an easier acceptance of foods considered healthy of local origin is laudable.

The introduction is clear and provides the information necessary to understand the objectives of the study.

The methodology is adequately structured and supported by adequate visual schemes i.e. flow-chart, flow diagram and figure, useful for quickly understanding some fundamental steps of the study. The various activities carried out are adequately presented.

Results are consistent with the study objective and, in my humble opinion, are satisfactory and well-explained both in descriptive terms and visual. This presentation provides information that permits us to easily understand what has been done and achieved.

The discussion is in line with the results and a consistent comparison with the literature is allowed for.

Moreover, I appreciated the concise conclusions which also recall the fundamental function of parents. Maybe, limitations could be extended.

Best regards.

Author Response

To Reviewer: Please see the attachment.
